# Automated Systems for Estrous and Calving Detection in Dairy Cattle

**Camila Alves dos Santos** [1,*] **, Nailson Martins Dantas Landim** [2] **, Humberto Xavier de Araújo** [2] **and Tiago do Prado Paim** [1]

[1] Instituto Federal de Educação, Ciência e Tecnologia Goiano, Rodovia Sul Goiana, Km 01, Zona Rural, Rio Verde 75901-970, Brazil; tiago.paim@ifgoiano.edu.br

[2] Institute of Education, Science and Technology of Tocantins, Universidade Federal do Tocantins, Av. NS-15, ALCNO-14, Plano Diretor Norte, Palmas 77001-090, Brazil; mac@uft.edu.br (N.M.D.L.); hxaraujo@uft.edu.br (H.X.d.A.)

[*] Correspondence: camilaalvesdossantos240@gmail.com

**Abstract:** Purpose: The objective of this review is to describe the main technologies (automated activity monitors) available commercially and under research for the detection of estrus and calving alerts in dairy cattle. Sources: The data for the elaboration of the literature review were obtained from searches on the Google Scholar platform. This search was performed using the following keywords: reproduction, dairy cows, estrus detection and parturition, electronic devices. After the search, the articles found with a title related to the objective of the review were read in full. Finally, the specific articles chosen to be reported in the review were selected according to the method of identification of estrus and parturition, seeking to represent the different devices and technologies already studied for both estrus and parturition identification. Synthesis: Precision livestock farming seeks to obtain a variety of information through hardware and software that can be used to improve herd management and optimize animal yield. Visual observation for estrus detection and calving is an activity that requires labor and time, which is an increasingly difficult resource due to several others farm management activities. In this way, automated estrous and calving monitoring devices can increase animal productivity with less labor, when applied correctly. The main devices available currently are based on accelerometers, pedometers and inclinometers that are attached to animals in a wearable way. Some research efforts have been made in image analysis to obtain this information with non-wearable devices. Conclusion and applications: Efficient wearable devices to monitor cows' behavior and detect estrous and calving are available on the market. There is demand for low cost with easy scalable technology, as the use of computer vision systems with image recording. With technology is possible to have a better reproductive management, and thus increase efficiency.

**Keywords:** precision livestock farming; livestock; management; milk; reproduction

## 1. Introduction

The demand for milk in the world is expected to increase by 58% by 2050, which requires a substantial increase in current production levels [1]. To address it, better dairy production management practices are imperative in order to provide better nutrition, reproduction, health, animal welfare and social and environmental conditions [2]. In the search for greater efficiency, precision livestock farming (PLF) through hardware and software seeks to obtain data from many production stages, which are used to improve farm management and optimize the returns per animal [3].

Cattle reproduction efficiency is highly correlated with profitability. The inadequate reproductive performance manifested with prolonged calving intervals and increased involuntary slaughter, or both, may result in less milk production, fewer calves per cow's lifetime and increased replacement costs [4]. These factors lead to a lower financial return

due to a decrease in the reproductive indices of the herd [5]. To perform a good reproductive management, it is necessary to adjust the monitoring and detection of signals, which will be indicators of the moment of insemination used by a cow [6]. Visual observation demands increased investment of time and manpower and to make estrous observation more efficient, automatic detection technologies are being used on farms [7]. In this way, the objective of this review is to show the state of the art of estrous and calving detection systems in dairy cattle, presenting commercially available systems and publications envisioning the development of new systems.

## 2. Estrous Detection Systems

Estrus is defined as the period when the female is receptive to mating with a mature male. In other words, it is the moment when the ovaries release the oocytes and the female can become pregnant [8].

During the estrous period, females undergo behavioral changes due to changes in the progesterone and estrogen levels [9]. The most pronounced signs of estrus is that the cow is standing while being mounted, which occurs $28.7 \pm 5.3$ h before ovulation [10]. During this period, cows stand to be mounted by other cows or move forward slightly with the weight of the mounting cow. The average duration of standing behavior is 15 to 18 h, but estrous duration may vary from 8 to 30 h among cows. A cow in estrus usually stands to be mounted between 20 and 55 times [11]. Each mount lasts three to seven seconds. The acceptance of the mount is most commonly expressed in the night period (6:00 pm–6:00 am), with up to 68% of the mounting events happening at night [12].

Secondary signs of estrus varies in duration and intensity and are related to: up to 4-fold increase in movement in the pen; reduction in feed and water consumption; attempt to mount other cows, mucus discharge, sniffing the genitalia of other cows, swelling and reddening of the vulva, chin resting and back rubbing of another cow and increased vocalization [13]. Generally, cows at the beginning of estrus are more likely to mount estrous cows [14]. Estrous expression is increased with a greater interaction and stimulation of other animals simultaneously in estrus [15]. When estrous behavior is detected by visual observation, the cow should be inseminated 4–12 h after mounting acceptance, so when the cow stands to be mounted in the morning it should be inseminated in the afternoon and, if it is detected in the late afternoon, the cow should be inseminated in the morning [16].

There are many factors that affect estrous expression in dairy cattle [17]. The type of housing is a factor that can influence the estrous behavior, as, due to concrete floors, cows have reduced mounting behavior. This fact can be explained by floor slippering, which can decrease the frequency of mounting behavior [8]. Animals confined in paddocks receiving a diet in the feeder may present a greater intensity in estrous behavior due to a higher concentration of females in a smaller area and less time spent with grazing and rumination [14].

The changes in the production systems in the last 30 years led to a lower expression of mounting behavior [18]. High production cows may have a reduction in their estrous behavior compared to nulliparous heifers due to the higher rate of estrogen metabolism, which is the hormone responsible for triggering estrous behavior [19]. Thus, visual direct observation of estrus has become less effective, requiring auxiliary methods for its identification [20]. The use of automated systems that assist in estrus detection, hormonal synchronization and computerized record systems are ways to improve the reproductive indexes of dairy cattle [21].

The use of auxiliary methods for the identification of estrus, such as tail chalk, when conducted well, can reach insemination rates of around 97% in heifers. A limitation of this technique is false positives, close to 5%, which are caused by the act of licking or rubbing [22]. The development of automated technology that can accurately identify the beginning of estrous behavior also opens new doors for the for adjusting the insemination schedule [23,24]. An example is the use of female sexed semen, as, due to the sex sorting process, the sperm survive for a shorter time in the female's reproductive tract; thus, fine-

tuning the cow's insemination schedule according to the ovulation schedule can lead to gains in pregnancy rates. The intensification of sexed semen use with better pregnancy rates can have an economic impact on dairy farming [25].

The increase in herd size, workload and physiological aspects, such as the duration of the estrous period, are also limiting factors for the visual detection of estrus. More periods of visual observation of the herd would be necessary. Thus, automated detection systems (Table 1), such as activity monitors using accelerometers and acoustic sensors (necklace, earrings, pedometers, pressure and friction detectors), are already used in large-scale dairy production. The limitation of wearable devices is that movement and temperature detection can be interfered by environmental or physiological disturbances [26–28].

**Table 1.** Commercially available estrus detection systems.

| Location | Technology/ Trade Name | Parameters | Link |
|---|---|---|---|
| Lower member | AfiAct Pedometer Plus® (Afimilk, Kibutz, afikim, Israel); Nedap Realtime Leg ® (Nedap Livestock, Management, The Netherlands, Marketed as CowScout S leg, GEA farm, Technologies GmbH, Bönen, Germany); Track a Cow® (ENGS Systems Innovative Dairy Solutions, Rosh Pina, Israel); Ice Qube® (IceRobotics Ltd., Edinburgh, UK) | Activity (steps) Lying time (min) | https://www.afimilk.com accessed on 5 February 2021 https://www.gea.com accessed on 5 February 2021 http://www.trackacow.co.uk accessed on 5 March 2021 https://www.icerobotics.com/cowalert accessed on 5 March 2021 |
| Ear | Cow Manager SensoOr (Agis Automatisering, Harmelen, The Netherlands) | Ruminating time (min) Active time (min) Time not active (min) | https://www.cowmanager.com accessed on 5 June 2021 |
| Ear | SmartBow® (smatbow GmbH, Jutogasse, Austria) | Animal Activity | https://www.smartbow.com/ accessed on 5 June 2021 |
| Sacral region | Kamar® (Kamar Inc., Zionsville, IN, USA) | Pression | http://www.kamarinc.com/ accessed on 5 June 2021 |
| Sacral region | Estrotec® (RockwayInc., Spring Valley, WI, USA) | Friction | https://estrotect.com/ accessed on 5 June 2021 |
| Sacral region (eletronic) | Heat-Watch® (DDx Inc., Denver, CO, USA) | Pression | https://www.cowchips.net/ accessed on 5 July 2021 |
| Neck | HR Tag® (SCR Engineers Ltd., Netanya, Israel) | Neck movement | https://www.nmr.co.uk accessed on 5 July 2021 |
| Reticulum Rumen | SmaXtec® (SmaXtec animal care GmbH, Austria) | Rumen Movement | https://smaxtec.com/en/ accessed on 5 July 2021 |

## 2.1. Pedometers and Accelerometers

Pedometers attached to the cow's leg are devices that record the number of steps taken per unit of time [11]. This number is used as indicator of walking activity, which is markedly increased during the proestrus and estrus of dairy cows. Accelerometers are tags that can be placed on a collar around the animal's neck. They identify the activity of the cow related to the upward movements of the head and neck during walking and mounting behavior, being transformed into an index that represents the weighted standard deviation of each basal activity of the cow itself [17,29].

Silper et al. (2015) [17] analyzed two commercial estrus detection systems, an accelerometer attached to a collar, HeaTime® (SCR Engineers Ltd., Netanya, Israel), and a pedometer attached to the cow's leg, IceTag® (Ice Robotics Ltd., Edinburgh, UK), and obtained a positive predictive value of 84.7% and 98.7%, respectively. The results showed that the two systems identified estrus accurately and with correlated characterization and times.

Schwenizer et al. (2019) [30] evaluated the estrus detection device using a 3D accelerometer attached to an ear tag (SMARTBOW®, Smartbow GmBH, Weinberg, Austria) in Holstein cows. These authors observed a sensitivity rate of 97% and specificity of 98%, demonstrating that the system is suitable for the detection of estrus in confined cows.

## 2.2. Radiotelemetry Pressure Detectors

Radiotelemetry systems are also commercially used to detect mounting activity through pressure detection. The HeatWatch (HeatWatch® DDx, Inc., Denver, CO, USA) consists of a mini radionic transmitter connected to a pressure sensor in a rigid plastic box embedded in a nylon package that is adhered to the tail hair in the sacral region of the cow [24]. The device is activated by the weight of the mounting animal during a minimum period of 2 s; then, the transmitter sends the breeding acceptance signal with the animal's identification to the system. In general, the analyzed performances of this device vary from 37% to 94% [31].

## 2.3. Alteration of Body Temperature

The female's body temperature increases by 0.4 °C during estrus; in this way, the reticulum–rumen boluses can be used to monitor the temperature variation of cows, being able to detect the moment of estrus [32]. Smaxtec® is an example of a device available on the market for this purpose (Table 1).

Mayo et al. (2019) [33] studied the efficiency of different estrous precision monitoring technologies in dairy cows (AfiAct Pedometer Plus®, Nedap Realtime Leg®, Track a Cow®, DVM Bolus®, Cow Manager SensOr®, HR Tag® and Ice Qube®) and demonstrated that all monitoring systems were able to detect estrus, being as effective as visual detection.

## 2.4. Imaging

Techniques based on computer vision with artificial intelligence can minimize the challenges mentioned above as the devices for acquisition and image processing devices will not be in direct contact with the animal. At the same time, if one camera can collect information from several animals, it can potentially decrease the cost per animal of the technology [20]. The monitoring of estrous behavior by video recording can be based on several methodologies (Table 2) that can be combined, such as the length of an object in the camera, the height difference of the animal (optical flow) due to the female's mount and the flight line being assembled [34]. The studies of estrous detection by images presented in Table 2 were chosen because they present the methodology and the results with rates of accuracy in the available tests.

**Table 2.** Results of detection of estrous cows through video recording and computer vision systems.

| Author | Tsai et al. (2014) [26] | Yang et al. (2017) [35] | Guo et al. (2019) [36] | Arago et al. (2020) [37] |
|---|---|---|---|---|
| **Breed** | Holstein | Holstein | Holstein | Holstein, Sahiwal |
| **Camera** | IP DOME | Infrared | Fixed IP | PTZ |
| **Algorithm** | Motion detector; Region targeting; Foreground targeting; BLOB analysis. | Motion detector; Region targeting; Foreground targeting. | BSCF; SVM; Geometric and Optical flow. | FR-CNN,;Tracking SSD |
| **Object detection accuracy** | - | - | 98.3% | 94% SSD 50% FR-CNN |
| **Estrous detection accuracy** [1] | 100% (TP) 0.33% (FP) | - | 90.9% (TP) 4.2% (FP) | 50% |
| **Limitation** | Only during the day | Shadows of the cows | Individual identification | Not suitable for cattle without identification |

[1] TP = True positive; FP = False positive.

Tsai et al. (2014) [26] proposed a method of estrous detection in cows housed in a shed based on the breeding behavior through the length of the object in the image frame; they first determined a region of interest in the set of images where there was a higher level of movement of animals. When the behavior of one cow following another (length of two

cows) for two seconds was detected in the image frame and then cow mounting (length of 1.5 cows) for two more seconds, an estrous signal was declared by the system. This system had up to 100% true positive of the 53 events observed.

Yang et al. (2017) [35] sought to complement the technique proposed by Tsai et al. (2020) of estrous detection through mounts at night, when the frequency of estrus is greater. In this low light condition, the shadow of the cows is difficult to be detected for image processing, but the authors suggest that the problem can be solved by the binary image method.

Guo et al. (2019) [36] identified the estrous behavior of cows in a cowshed through image processing using geometric and optical flow methods. These authors used a background subtraction algorithm with color and texture characteristics to improve the determination of body regions of the cows.

Arago et al. (2020) [37] sought to identify the estrous behavior through bounding boxes and neural networks to identify the animals responsible for the breeding. The results for estrous detection were satisfactory, with a detection efficiency of 50%. However, these authors reinforce the machine learning challenge to identify animals.

During the estrous period, cows may show a variation in their body temperature, so research has also sought to explore the use of thermographic cameras to associate the surface temperature of cows with the estrous period. Talkuder et al. (2014) [38] took photos of the vulva of dairy cows on pasture with a thermographic camera and obtained results that the highest temperature of the vulva was observed 24–72 h before ovulation, while the lowest temperature was 48 h before ovulation, presenting results of sensitivity of 75%, specificity of 57% and a positive predictive value of 69%. According to Mottram et al. (2016) [39], there are challenges to eliminate temperature variation not related to estrus, such as variations in ambient temperature and humidity.

## 3. Calving Detection Systems

Profitability in dairy farming is also influenced by the birth rate of live calves [40]. Despite the development of rearing systems, perinatal mortality (stillbirths) is still very high in confined systems ranging from 5 to 9.6% [41]. The costly effects of stillbirth losses are even greater when considering the effects on cow's lactation, causing losses due the increased risk of developing metritis and retained placenta, risk of primiparous not conceiving after the second service and lesser chances of conception compared to the rest of the herd, in addition to increased rates of discard for low milk production and inadequate reproductive performance [42].

Dystocia can be defined as difficult calving, which results in prolonged calving with human-assisted calf extraction. The videntification of animals in labor allows producers to assist in the event of dystocia. Cows that take more than 70 min to calve from the appearance of the amniotic sac outside the vulva have a greater risk of dystocia, and if delivery assistance is provided on time, this risk and the stress associated with this problem can be reduced [43,44]. Premature assistance to cow calving can negatively impair the duration of the calving process, the mother's health status and the vitality of the offspring [45].

Cows close to calving have lower body temperatures, relaxation of the pelvic ligaments, decreased feeding pattern and rumination, increased frequency of lying down and more frequently movement of head towards flank (2 h before delivery) [46]. Several technologies have been proposed for automatic calving prediction, most of which are based on wearable sensors, such as accelerometers, microphones detecting rumination activity, electromyography tags and temperature sensors [47]. However, these monitors must be connected to each cow, which can become costly and cause discomfort to the animal [48].

Zehner et al. (2018) [49] analyzed the digestive changes of cows near parturition through a muzzle sensor (RumiWatch®, Agroscope Etthenausen, Switzerland and Itin + Hoch GmBH, Liestal, Switzerland). These authors observed that the sensitivity and specificity of the model were satisfactory to detect one hour before parturition start; however, the positive predictive value was low and the false positives were considerably high. The tech-

nical advantages of using this type of device for calving detection are to take advantage of a device that is already in use for monitoring the health of cows, for calving detection [47].

Benaissa et al. (2020) [50], when studying the detection of calving in confined dairy cows through the combination of accelerometers (mounted on the neck and leg of the cows) and Ultraload Band (UWB) sensors, demonstrated that the combined systems increase the birth detection capability, with an accuracy of 72–87% and sensitivity of 63–85%. This study demonstrated the potential use of the devices in a multifunctional way, but more specific studies need to be carried out to validate the technology for delivery prediction.

VelPhone® (Medria, Chateaugiron, France) is an intravaginal birthing detection device based on the decrease in the vaginal temperature close to calving and the temperature difference when the device is expelled from the vagina moments before parturition. This system presented positive value predictors of up to 100% [51].

Jensen (2012) [52] studied the behavior of dairy cows housed in individual pens in the prepartum period by associating video analysis and accelerometers attached to the cows' legs 96 h before calving. It was shown that the cows became more active, having more movements of lying down and getting up, the day before delivery than in the 2–4 days before delivery.

Cangar et al. (2008) [53] monitored the behavior of cows through video analysis 24 h before calving. These authors based the analysis on: the coordinates of the view from the central point of the top view of the cow; the walking path; distance covered; orientation of the main axis; body width/length ratio; hip length; and back area to determine specific behaviors, such as standing, eating or drinking. The behavior of standing or lying down (75%) and the behavior of eating and drinking (87%) were correctly classified.

Monitoring pre-calving cows using video cameras and computer vision techniques is still rare, but has the potential to predict calving. Sumi et al. (2018) [48] proposed a system for automatic detection of calving behavior using a 3D camera along with an image processing system with background subtraction, allowing the detection of the cow's posture change every 1 h.

## 4. Conclusions

Currently, the main systems available for detection of estrus and parturition are the inclinometer, accelerometer and pedometer, where wearable devices are required. The use of technologies can help the control of zootenic data on the farm, optimizing the use of labor. Additionally, when well used, they can provide fees that help the producer in decision-making on the property. The development of detection methods based on video recording and computer vision systems is promising, as there is no need to attach individual items per animal, allowing for lower costs per animal. Thus, it is crucial to adapt each technology to the reality of the property in which it will be applied.

**Author Contributions:** Conceptualization, T.d.P.P. and C.A.d.S.; writing—original draft preparation, C.A.d.S. and T.d.P.P.; writing—review and editing, N.M.D.L. Reviewing final draft, N.M.D.L. and H.X.d.A.; funding acquisition T.d.P.P. and H.X.d.A. All authors have read and agreed to the published version of the manuscript.

**Funding:** This review was funded by Instituto Federal de Ciência, Educação e Tecnologia Goiano and the Universidade Federal do Tocantins.

**Acknowledgments:** Thanks to the Instituto Federal de Ciência, Educação e Tecnologia Goiano and the Universidade Federal do Tocantins for the financial support.

**Conflicts of Interest:** The authors declare no conflict of interest.

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
