# Peer review of "Automated Systems for Estrous and Calving Detection in Dairy Cattle"

_agriengineering, doi:10.3390/agriengineering4020031_

Round 1

Reviewer 1 Report

The review article entitled "Automated systems for estrous and calving detection in dairy cattle’’ aimed to review the main technologies available commercially and under research looking for automated estrous and calving detection in dairy cattle. The review is written in an easy way for the reader. But there are some minor issues, please correct them so that the review becomes acceptable for publication:

- Please follow the guidelines of the journal for the reference style.

- There are some references not found in the reference list:

  • Line31: Ruviano et al., 2020.
  • Line34: Costa et al., 2018.
  • Line 42: Sewalen et al., 2008.
  • Line 89: Nebel et al.,
  • Line 110: Silper et al., 2017.
  • Line 133: Vicentini et al., 2018.
  • Line 155: Tsai et al (2020).
  • Line 159: Guo et al (2019).
  • Line 201: Zeherner et al (2019).
  • Line 179: Óto Szenci et al., 2017.

3- Please correct the spelling of “mouting” in lines 122, 126 and 152 to “mounting”

4- Please complete the details of reference 1 Line 247 “ J. Adv. Comput. Sci. Appl11 (2020): 303-311”

5- Line 281-283: Please delete “Larson, L. L., & Ball, P. J. H. (1992). Regulation of estrous cycles in dairy cattle: A review. Theriogenology, 38(2), 255–267. doi:10.1016/0093-691x(92)90234-i

6- Line 285: Please complete the details of reference “56 (4): 673-679.”

7- Line 300: Please complete the details of reference “ 308-313”

8- Line 310: reference No. 27 not found in the text.

9- Line 328: please complete the details of reference “130: 19-25.”

Reviewer 2 Report

Review – Automated systems for estrous and calving detection in dairy cattle.

The manuscript is a review of the usage of technology to detect estrus and calving in dairy cattle. The topic is relevant to the dairy industry as technology have increased over the years in dairy herds.

Major comments

The manuscript is hard to follow, as there are many English typos. Considering that is a literature review of the topic, is expected that the authors explore more the literature already published. There is a lack of citations (authors don’t use references after certain statements) and some of the citations are not used corrected (studies cited did not evaluated statement written by the authors).

In the introduction is important to state why the literature review is important, what are the points lacking on the literature? How the technology can be used to improve dairy management and perhaps improve herd performance (production and reproduction). On the section “estrous detection systems” the authors should describe what is estrus, why estrus is important for good reproduction performance, then state what is the issue (cows producing greater volume of milk, labor, housing systems, etc, have reduced estrus behavior). Why were sensors created?  The calving section is much shorter than the estrus section.

Table 1 and 2 should have more information about the sensors. What are they used for? What is the specificity and sensitivity of them? Add the references for each item you describe on the table. Furthermore, tables should stand alone, make sure that all the information that is needed is described on the title of the table or legend.

Line 11 – 12 – Rephrase “The objective of this review is to describe the main technologies (automated activity monitors) available commercially and under research for detection of estrus and calving alerts in dairy cattle”.

Line 12 – What was the criteria for selecting specific online journals?

Line 14 – Avoid starting sentences with acronym.

Line 15 – “it could be used to improve herd management and…” .

Line 16 – Visual observation for estrus detection and calving, are activities that require labor and time…

Line 18 – this is not true. The sensor has the potential to help producers to identify estrous and calving alert, but it has to be used correctly. Sensor data will only be valuable if transformed into information that is useful for decision making. The adoption of precision dairy monitoring can improve or maintain animal welfare on dairy herds.

Line 23 – “However”, as a parenthetical remark usually should be placed within a phrase, not at the beginning of a sentence.

Line 25 – With technology is possible to have better reproductive management, and thus increase efficiency. Re-word to clarify your conclusion.

Introduction

Line 30 – 34 - Change to “The demand for milk in the world is expected to increase by 58% by 2050, which requires a substantial increase in current production level (Ruviano et al., 2020). To address it, better herd management practices are needed (add a reference)”. Costa et al. 2018 is not in your citation list. If the reference is a review, you have to find the actual papers that have studied those factors.

Line 35 – What the authors mean by “data from nay sources”? Be clear.

Line 38 – add a reference to back up your point.

Line 39 – replace “birth” to “calving”.

Line 30 – 40 – increased involuntary culling

Line 40 – what is a “calf crop” – do you mean fewer calves per lifetime of the cow’s lifetime? Re-phrase.

Estrous Detection Systems

Line 47 – define estrus before the behavior changes.

Line 47/48 – elevated concentration of estradiol. Remove low progesterone, what is low progesterone?  Cows with incomplete luteolysis (higher levels of progesterone) still show estrus behavior. You can rephrase to say that estradiol and progesterone play an important role in the estrous cycle and thus it can impact estrus behavior.

Line 49 – signs of estrus

Line 50 – which occurs around 30 hs

Line 52 – Standing behavior

Line 52 – Replace “heat” to “estrus” and add a citation

Line 53 – A cow in estrus normally stands to be mounted…. remove “during her estrous period” and add a reference.

Line 54 – What is the relevance of adding this sentence to your review? Make sure that older citations come first than newer.

Line 57 – Secondary signs of estrus

Line 57/58 – what do you mean by “up to 4-fold increase”? Increase in what movement?

Line 59 – add reference

Line 61 – estrus - the definition of Estrus noun “The cow displays estrus approximately every 21 days”; and the definition of Estrous adjective “The intensity of estrous expression can be measure using sensors”. Change estrus/estrous accordingly throughout the manuscript.

Line 66 – fix the reference order

Line 68 – there is no link from the previous paragraph to this one.

Line 68 – There are many factors that affect estrous expression in dairy cattle (references). The type of housing is a factor that can influence the estrus behavior, due to concrete floors cows have reduced mounting behavior.

Line 72 – What is the relevance of adding this portion “diet in the feeder”.

Line 75 – add a reference

Line 79 – this is wrong citation; those researchers did not studied clearance of hormones due to high metabolism. Please, add the correct reference.

Line 79 – estrus

Line 79 – add a reference

Line 79 – add a sentence explaining that with clearance of hormones, cows have reduced the gold standard behavior of estrus, standing to be mounted.

Line 85 – 87 – Is that false positive in heifers??

Line 86 – licking or rubbing the chalk – describe as tail chalk and not tail paint.

Line 87 – automated technology and not techniques

Line 88 – what type of behavior are you describing here? Is it primary or secondary? What type of sensor? Pedometers, accelerometers?

Line 89 – Can you explain “insemination schedule”?

Line 96 – estrus

Line 96 – detection of estrus

Line 104 – Reference

Line 105 – Add a reference for walking activity behavior, a pedometer can measure the number of steps, but not necessarily walking activity. Cows normally give a stationary step, which increase the number of steps, but not walking.

Line 121 – What is correlation between radiotelemetry and pedometers/accelerometers? On Line 79 the authors described that cow had reduce steroidal hormones that trigger estrus behavior and that visual estrus has become less effective (due to cows showing less behavior), so why do you recommend a sensor that relies on mounting behavior?

Line 130 – The authors are describing ruminal temperature? Rectal or vaginal temperature? What are the reasons that body temperature increases by 0.4 degrees?

Line 135 – Mayo et al 2019 describe overall sensors, not only body temperature, but this sentence does also not fit in this section.

Line 143 – add a reference

Line 148 – what the authors mean by “breeding behavior”?

Line 153 – add the specificity and sensitivity of this methodology

Line 159 – Cowshed? Dairy barn?

Line 164 – re-write this sentence, isn’t clear what the authors meant.

Calving Detection Systems

Line 177 – add a reference

Line 193 – 200 – The authors described technologies that can be useful for calving detection. As this is a review, describe each of those sensors, pros and cons.

Conclusion – Re-write the conclusion, it does not reflect the main topics discussed in the text.

Table 1. What is the relevance of adding this table to your review? Add columns describing sensitivity and specificity for estrus detection and calving. Add another column with the references

For the boluses, why the authors describe one as “reticulum rumen” and the other one only as “rumen”?

Table 2. for the decimals, use “.” not “,”. Table should stand alone, be more specific. Why the authors chose only those 4 studies to present in the table?

References – there are some references missing and some typos.

Round 2

Reviewer 2 Report

The authors have done major changes in the manuscript and it reads well now. Thank you.

My main concern is with the selection of the manuscripts used in this review. Authors have to make it clear how the the manuscripts were chosen. See my comments bellow

Line 13-14: “Sources: Data for the preparation of the literature review were obtained from searches in online journals available for consultation involving reproduction and precision livestock farming (PLF)”.

Authors must justify how this selection was performed. As for any other literature review, you should include what was the words used for search, what was the criteria to choose specific papers (did you choose by title, abstract, relevance, number of citations, etc.).

Line 132: Standing behavior not “standing hear”

Line 133: “A cow in estrus”.

Line 137: estrus

Line 138: replace shed with “pen”.

Line 141: “estrus” not “estrous” estrus - the definition of Estrus noun “The cow displays estrus approximately every 21 days”; and the definition of Estrous adjective “The intensity of estrous expression can be measure using sensors”. Change estrus/estrous accordingly throughout the manuscript.

There are more than 4 manuscripts published using video recording and computer vision systems in the literature. Make sure that, in the text, you justify why you choose those specific 4 manuscripts.

DVM bolus system doesn’t exist anymore, remove from the table. The authors may find another system that collect the same information.

Round 3

Reviewer 2 Report

Thank you for adding the information required.